# Clonal Myeloproliferative Disorders in Patients with Down Syndrome—Treatment and Outcome Results from an Institution in Argentina

**DOI:** 10.3390/cancers14133286

**Published:** 2022-07-05

**Authors:** Carla L. Pennella, Tamara Muñoz Cassina, Jorge G. Rossi, Edgardo M. Baialardo, Patricia Rubio, María A. Deu, Luisina Peruzzo, Myriam R. Guitter, Cristian G. Sanchez de La Rosa, Elizabeth M. Alfaro, María S. Felice

**Affiliations:** 1Department of Hematology-Oncology, Hospital de Pediatría S.A.M.IC. “Prof. Dr. Juan P. Garrahan”, Combate de los Pozos 1881, Buenos Aires C 1245 AAM, Argentina; tamaracassina@gmail.com (T.M.C.); patrirubio13@gmail.com (P.R.); alejandradeu86@gmail.com (M.A.D.); luisinaperuzzo@gmail.com (L.P.); myriamguitter@yahoo.com (M.R.G.); cgsdoc@yahoo.com.ar (C.G.S.d.L.R.); elizabeth2006_alfaro@yahoo.com.ar (E.M.A.); marisa.felice2013@gmail.com (M.S.F.); 2Department of Immunology and Rheumatology, Hospital de Pediatría S.A.M.I.C. “Prof. Dr. Juan P. Garrahan”, Combate de los Pozos 1881, Buenos Aires C 1245 AAM, Argentina; jrossi@garrahan.gov.ar; 3Department of Genetics, Hospital de Pediatría S.A.M.I.C. “Prof. Dr. Juan P. Garrahan”, Combate de los Pozos 1881, Buenos Aires C 1245 AAM, Argentina; baiaed@yahoo.com.ar

**Keywords:** Down Syndrome, acute myeloid leukemia, transient abnormal myelopoiesis, megakaryoblastic leukemia, chemotherapy, outcome research, pediatric hematology/oncology

## Abstract

**Simple Summary:**

Around 30% of children with Down Syndrome (DS) will develop Transient Abnormal Myelopoiesis (TAM) and 20% of them will progress to Acute Myeloid Leukemia (AML), mostly Megakaryoblastic Leukemia (AMKL). The optimal balance between treatment intensity and treatment-related toxicity has not yet been defined; neither the prognostic factors that determine the risk of developing AML nor the outcome. The aims of our retrospective study were to analyze the demographic/biological features of this population, identify possible risk factors and the optimal treatment. We observed that early intervention in TAM is effective to prevent a dismal outcome. The strongest poor-prognostic factor of DS-AML was sporadic DS-AML (non-AMKL immunophenotype), as well as complex karyotype and young age. Classical Myeloid Leukemia associated with DS (ML-DS) good outcome is mainly related to their low relapse rate. Even though the augmented sensitivity to chemotherapy seen in DS patients must be kept in mind, our data do not support the omission of high doses of cytarabine in ML-DS.

**Abstract:**

Children with Down syndrome (DS) are at an increased risk of developing clonal myeloproliferative disorders. The balance between treatment intensity and treatment-related toxicity has not yet been defined. We analyzed this population to identify risk factors and optimal treatment. This single-center retrospective study included 78 DS patients <16 years-old with Transient Abnormal Myelopoiesis (TAM, *n* = 25), Acute Myeloblastic Leukemia (DS-AML, *n* = 41) of which 35 had classical Myeloid Leukemia associated with DS (ML-DS) with megakaryoblastic immunophenotype (AMKL) and 6 sporadic DS-AML (non-AMKL). Patients with DS-AML were treated according to four BFM-based protocols. Classical ML-DS vs. non-DS-AMKL were compared and the outcome of ML-DS was analyzed according to treatment intensity. Only four patients with TAM required cytoreduction with a 5-year Event-Free Survival probability (EFSp) of 74.4 (±9.1)%. DS-AML treatment-related deaths were due to infections, with a 5-year EFSp of 60.6 (±8.2)%. Megakaryoblastic immunophenotype was the strongest good-prognostic factor in univariate and multivariate analysis (*p* = 0.000). When compared ML-DS with non-DS-AMKL, a better outcome was associated with a lower relapse rate (*p* = 0.0002). Analysis of administered treatment was done on 32/33 ML-DS patients who achieved CR according to receiving or not high-dose ARA-C block (HDARA-C), and no difference in 5-year EFSp was observed (*p* = 0.172). TAM rarely required treatment and when severe manifestations occurred, early intervention was effective. DS-AML good outcome was associated with AMKL with a low relapse-rate. Even if treatment-related mortality is still high, our data do not support the omission of HDARA-C in ML-DS since we observed a trend to detect a higher relapse rate in the arm without HDARA-C.

## 1. Introduction

Trisomy of chromosome 21 (cr21) is the most common aneuploidy in live births and is the cause of Down Syndrome (DS). Approximately 10–30% of neonates with DS develop a unique pre-leukemic condition called Transient Abnormal Myelopoiesis (TAM) [1,2,3,4,5,6,7,8,9], with circulating blasts that have morphologic and immunologic features of the megakaryocytic lineage [1,2,3,4,5,6,9]. A multistep process of leukemogenesis has been postulated in which trisomy 21 represents the “initiating” event, impairing in utero erythro-megakaryopoiesis. Thus, progenitor cells within the fetal liver are primed for the acquisition of somatic N-terminal truncating *GATA1* mutations that reflect a ‘‘secondary hit”’, giving place to the occurrence of TAM [1,2,3,4,6,8,9,10,11,12,13] at a median age of 3–7 days of life [1,2,3,4,6,8]. Additionally, these two genetic alterations cooperate to increase the number of active cytarabine metabolites present in DS blast cells, enhancing cytotoxicity [10,14].

After birth, hematopoiesis naturally transitions from the fetal liver to the bone marrow and the *GATA1* megakaryoblastic clone becomes quiescent. Most patients (70–80%) have spontaneous regression of TAM within a few months without intervention needed [1,2,3,4,5,6]. However, the overall mortality is 10–20% [1,2,3,4,5,6,8]. Multiple clinical features associated with increased risk of morbidity and mortality have been described [2,3,4,5,6,8,9]: high WBC count (>50,000–100,000 leukocytes/mm^3^), thrombocytopenia (platelet count <100,000/mm^3^), signs of cholestasis or liver dysfunction, bleeding diatheses, heart or renal failure, effusions (pleural, pericardial, ascites or hydrops), preterm delivery, low birth weight (<3 Kg) and failure of spontaneous remission [15,16,17,18,19]. Though no optimal treatment strategy or intervention criteria are established, there is a consensus that when one of these clinical features are present, early treatment is recommended [5,9].

Approximately 20% of children with TAM progress to Acute Myeloid Leukemia (AML) [1,2,3,4,5,6,8,9,11] when persistent *GATA1* mutant cells acquire additional mutations, most frequently in genes encoding members of the cohesin protein family, epigenetic regulators and signaling molecules [20,21]. Approximately 80% of the cases are classified as Megakaryoblastic Leukemia (AMKL), and its incidence increases by 500-fold in children with DS by the age of 4 years compared to the chromosomally normal population [3,8,9,10,11]. Until now, no predictive factors regarding when TAM will develop into AML have been clearly recognized [2], although some were identified: additional chromosome abnormalities at the TAM phase [15,22], pleural effusion and thrombocytopenia at diagnosis [16], longer time for regression of TAM [17] and low expression level of *GATA1* mutated [19]. The TAM Prevention 2007 (TMD07) trial aimed to reduce the incidence of AML in children diagnosed with TAM by applying a low-dose cytarabine treatment in combination with minimal residual disease (MRD) monitoring to eradicate the *GATA1*-mutated clone [23]. This study, as other trials, failed to demonstrate that early chemotherapeutic intervention could prevent progression to DS-AML [16,17].

Consequently, DS patients comprise 15% of pediatric AML cases enrolled on clinical protocols [24] and classical Myeloid Leukemia associated with DS (ML-DS) constitute a separate entity defined by: somatic *GATA1* mutations, younger age (mainly < 5 years old), blasts with not only megakaryocytic cell-surface markers but also erythroid markers and CD7 expression [25,26,27], superior clinical outcome, lack of Central Nervous System (CNS) involvement, and increased sensitivity to chemotherapeutic agents as well as increased treatment-related toxicity [6,8,9,10,11]. Children older than 5 years of age may not have *GATA1* mutations and such cases should be considered as “conventional” Myelodysplastic Syndrome or AML [28]. The optimal balance between dose intensity and the risk of treatment-related toxicity has not yet been defined. Multiple study groups have tried to find risk factors for better treatment adequation, and poor early response, gain of chromosome 8 [29], chromosome 7 monosomy [30], isolated +21c [31], age > 2–3 years [31,32], extramedullary invasion at initial diagnosis [33], MRD positivity on day 28 of induction [34] and antecedent of TAM [16] have been recognized as such.

There is a consensus that Hematopoietic Stem Cell Transplantation (HSCT) should only be performed as part of a second-line treatment if necessary [6,10,24,35,36]. Concern about anthracycline exposure in patients with DS is well justified and reduced doses are recommended [14,24,29,30,34,37,38,39]. Less treatment-related mortality (TRM) was described when more time is given for marrow recovery [35,37,40]. Several treatment groups have identified incorporation of High-Dose Cytarabine (HDARA-C) as responsible for better outcomes in DS-AML [27,31,37,41]. In the ML-BFM98 study [42], the cumulative doses of anthracyclines and cytarabine were reduced compared to the BFM93 study omitting the HAM block, and high cure rates were achieved. That study supported the use of HDARA-C and proposed further reduction of cytarabine doses from 3 g/m^2^ to 1 g/m^2^. Japan Children’s Cancer and Leukemia Study Group AML 9805 Down Study [33] achieved a 5-year-Event Free Survival (EFS) of 82.6 (7.9)% and concluded that the dosage of 1 g/m^2^ may be sufficient. As well, in the NOPHO-AML88 and NOPHO-AML93 [40] patients that received reduced therapy (an average of 67% of the cytarabine dose) had an EFS of 92%, comparable to the 76% EFS in the group receiving full AML treatment (*p* = 0.17), being nevertheless all treatment failures due to relapses. The ML-DS 2006 protocol [29] proposed a further reduction of etoposide, CNS prophylaxis and excluded maintenance therapy keeping HDARA-C, without increasing relapse rate compared to the historical BFM-98 protocol.

On the other hand, some studies support the use of low-dose of ARA-C. The Japanese trials [30,32,43] achieved comparable EFS (80 (7)%, 83.3 (9.1)% and 85.5 (4.2)%) without HD-ARAC. The Hospital for Sick Children [44] observed similar EFS in patients treated with the low-dose and standard-dose of ARA-C (66.7% and 75%, *p* = 0.5, respectively), even though all five deaths in the low-dose group were because of the disease.

The aims of this study were to analyze myeloproliferative clonal disorders in the DS population, identify possible poor prognostic factors of DS-AML and evaluate the outcome according to treatment intensity. We found that not age, but megakaryoblastic immunophenotype was the strongest prognostic factor for DS-AML. Even if classical ML-DS (AML-DS with AMKL immunophenotype) good outcome is related to low relapse risk and despite their increased treatment-related toxicity, our data does not support the omission of high-doses of cytarabine as a trend to achieve a better outcome and no relapses were observed in the group who received HDARA-C.

## 2. Materials and Methods

This was a retrospective study performed at a tertiary care Pediatric Center in Argentina. From January-1990 to September-2020, all patients with a diagnosis of TAM and DS-AML under 16-years old were registered. Patients with DS-AML as well as non-DS-AML were enrolled in one of four consecutive treatment protocols inspired in BFM protocols: 1-AML90-BFM/HPG, 1-AML95-BFM/HPG, 4-AML99-BFM/HPG, 12-AML07-BFM/HPG. Details of treatment schemes have been described in previous reports [45,46], and is detailed in Figure 1. Briefly, the first block of the induction phase was the same for the four studies, consisting of an 8-day block with cytarabine–idarubicin–etoposide (AIE block). After the AIE block, the 4-AML99-BFM/HPG and 12-AML07-BFM/HPG protocols included a second induction course with HDARA-C (3.0 g/m^2^ every 12 h) and mitoxantrone (HAM block). Following induction phase 1-AM90-BFM/HPG, 1-AML95BFM/HPG and 4-AML99-BFM/HPG, a 6-week consolidation-phase was delivered. The protocol 12-AML07-BFM/HPG administered two blocks of consolidation, including cytarabine plus idarubicin (AI-block) followed by cytarabine in combination with mitoxantrone (hAM-block). The intensification-phase included HDARA-C plus etoposide: Protocol 1-AML90-BFM/HPG included two intensification blocks for all patients; Protocol 1-AML95-BFM/HPG one intensification-block for low-risk or two blocks for high-risk cases; and protocols 4-AML99-BFM/HPG and 12-AML07-BFM/HPG administered one intensification-block. The four studies included a maintenance phase until completing 18 months from the date of diagnosis. Adapted doses according to DS specifications (2/3 dose reduction of anthracyclines on AIE-block and consolidation/AI blocks) and the recommendation to avoid HAM-block since the AML-BFM 98 trial and ulterior reduction doses based on individual tolerance were considered for each patient. For patients younger than 24 months, HDARA-C doses were reduced according to age. DS-AML were not considered for HSCT during first-line treatment. All protocols were approved by the local Ethics Committee and patient informed consent was obtained in accordance with the Declaration of Helsinki.

In all cases, diagnosis was based on morphological, cytochemical and immunological features of atypical cells found by light microscopy, flow cytometry, G-banding, reverse transcription polymerase chain reaction (RT-PCR) and FISH studies, according to standard techniques on local laboratories [28,47,48,49]. Surface and cytoplasmic marker analysis was performed on cells from patients’ bone marrow aspirates by four to eight flow-cytometry according to the previously described protocols [48]. Chromosome analyses of BM specimens were performed according to standard techniques. Criteria for clonality, chromosome identification, and karyotype designation were based on guidelines as defined by the International System for Human Cytogenetic Nomenclature [49]. RT-PCR studies for *RUNX1-RUNXT1*, *CBFB-MYH11*, and *KMT2A-MLLT3* fusion transcripts were performed as previously described [50,51] and were incorporated since December 2002. The analysis of alterations in *GATA1* and *FLT3* genes was performed by PCR and Sanger sequencing method [52,53]. The last follow-up update was in May-2021. Diagnosis criteria for TAM was primarily defined by the presence of a *GATA1* mutation in a neonate with DS or mosaic DS combined with an increased blast count (blast threshold of >10%) with morphological and immunological features of megakaryocytic lineage or clinical features suggestive of TAM [5,28]. The diagnosis of AML was based on the presence of >20% abnormal myeloblasts on bone marrow smears. For ML-DS also cases with less than 20% of blasts were considered and for the assessment of megakaryocytic lineage, at least 10% of the blast cells needed to be positive for one or more of the platelet-associated antigens (CD41, CD42 or CD61).

Clinical, demographic and biological features and outcome analyses were performed for all patients. TAM was analyzed as a separate group. A comparative analysis was performed between classical ML-DS (AML-DS with AMKL immunophenotype) and non-DS-AMKL registered in the same period of time. Finally, for the classical ML-DS group, a separate outcome analysis was performed according to intensity of treatment defined as the use or not of at least one block of chemotherapy with HDARA-C. The parameters assessed for the analysis of clinical and demographic characteristics were age, sex, white blood cells (WBC) count and extra-medullary involvement sites at the time of diagnosis. For biological characterization: immunophenotype, cytogenetics, molecular and recurrent genetic abnormalities were analyzed. Complex karyotype (CK) was defined as ≥three cytogenetic abnormalities, including ≥one structural abnormality.

The outcome was assessed considering morphologic response in the bone marrow (BM) defining M1, <5% of blasts; M2, ≥5% and <25%; M3, ≥25% of blasts. Complete remission (CR) was defined by the absence of symptoms and signs of leukemia, normal clinical performance status, undetectable leukemic blasts in peripheral blood and CSF, and M1-BM aspirate after peripheral hematopoietic regeneration (recovery of neutrophils > 500/mm^3^ and thrombocytes > 50.000/mm^3^) followed by AIE-block. Patients who achieved CR after a second chemotherapeutic course were defined as “late-responders”. “Non-response” was defined as the failure to achieve CR, whereas children who died before achieving CR were defined as “deaths-during induction” cases and those who died after achieving CR were defined as “deaths in CR”. Induction failure (non-response or late-responders), relapses, occurrence of second malignancies, death-during induction and deaths in CR were defined as events. EFS probability (EFSp) was defined as the time from diagnosis to any event, whichever occurs first, or time to last follow-up for patients without events. To analyze the Overall Survival probability (OSp), death by any cause was defined as an event. Treatment-related mortality (TRM) was defined as any death occurring at any time during treatment and not related to relapse or second malignancy. If death occurred during a clinically or microbiologically documented infection, it was considered to be due to infectious complications.

STATA 11 statistical software was used. The prognostic value for discrete variables was assessed with *χ*^2^ test. The log-rank test was used to compare outcomes for subgroups identified by prognostic factors (immunophenotype, cytogenetic abnormalities and CK for DS-AML; and HDARA-C for classical ML-DS). A Cox model (single step) on Event Free Survival probability (EFSp) was used for multivariate analysis of prognostic factors. EFSp, OSp and Cumulative Incidence of Relapse probability (CIRp) were analyzed by Kaplan-Meier calculation and compared with the log-rank test.

## 3. Results

From January-1990 to September-2020, 725 AML patients were admitted and 68 (9.4%) of them were DS: 26 TAM and 42 DS-AML. From 116 AMKL, 36 (31%) were DS and 80 (69%) non-DS. A total of 15 patients were excluded from the analysis: 1 TAM lost in follow-up, 1 DS-AML (AMKL) lost in follow-up, and 13 non-DS-AMKL (2 lost in follow-up, 4 due to previous treatment, 3 blastic crisis of chronic myeloid leukemia and 4 with a previous history of myelodysplastic syndrome).

### 3.1. TAM

The median age of presentation of TAM was 27 days of life (1–150 days of life). The mean initial WBC count was 64.250/mm^3^ (range: 5.870–260.210/mm^3^, 6 (24%) with hyperleukocytosis > 100.000/mm^3^) and the mean platelet count was 166.000/mm^3^ (range: 23.000–800.000/mm^3^, 14 (56%) with thrombocytopenia < 100.000/mm^3^). Almost all patients presented clinical hepatic compromise at diagnosis: 23/25 hepatic enlargement (92%), and one patient presented hepatic failure. Only one patient had skin compromise and one had a pericardiac effusion. Of the 25 patients analyzed, 15 had comorbidities: 12 cardiac, 3 hypothyroidism, 1 duodenal atresia and 1 renal abnormality. *GATA1* mutations were analyzed in 21 cases, resulting in 85.7% of them being positive. Cytogenetics analysis was performed on all patients, and 22 presented only +21c, one del cr19, one DS mosaicism (47,XY,+21c/46,XY) and one DS/Turner mosaicism (45X,0/47,XY,+21c).

Only four patients with TAM required cytoreduction, three due to hyperleukocytosis symptoms and one respiratory distress and hepatic failure. Neither presented a later event. The 5-year EFSp was 74.4 (standard error [SE] 9.1, 95% confidence interval [CI] 51.5–87.7)%, and OSp 82.4 (SE 8.1, 95% CI 59.3–93)%. One patient died secondary to an aspiration event previous CR. Three patients died on CR, 2 due to sepsis and one to cardiac failure. Three developed leukemia: two AMKL (leukemia free survival of 8 and 13 months) and one ALL (leukemia free survival of 69 months).

### 3.2. DS-AML

Table 1 summarizes the 41 DS-AML patients’ clinical characteristics at initial diagnosis. Only two patients had extra-medullary involvement at diagnosis in the CNS.

DS-AML cases were enrolled in four consecutive treatment protocols: 1-AML90-BFM/HPG (*n* = 9) 1-AML95-BFM/HPG (*n* = 5), 4-AML99-BFM/HPG (*n* = 7), and 12-AML07-BFM/HPG (*n* = 20). All 9 patients enrolled on 1-AML90-BFM/HPG and 2 on 1-AML95-BFM/HPG have been previously reported [54]. When compared TRM between 1-AML90-BFM/HPG/1-AML95-BFM/HPG vs. 4-AML99-BFM/HPG/12-AML07-BFM/HPG, a significant difference was observed: 50% vs. 12.5%, respectively (*p* = 0.006), with 28.5% deaths-during induction and 21.4% deaths in CR in the first two protocols vs. 3.7% and 7.4% in the latest

For the total population of DS-AML, the 5-years EFSp was 60.6 (SE 8.2, 95% CI 42.8–74.4)% (Figure 2) with a median follow-up of 45 (1–153) months. Patients with constitutional trisomy 21 alone showed similar outcomes to those with other cytogenetic abnormalities: 69.2 (SE 12.8, 95% CI 37.3–87.2)% vs. 64.3 (SE 10.2, CI 41–80.4)% (*p* = 0.952). However, when divided between those with CK and without, a significant difference was observed (*p* = 0.043) (Figure 2). Finally, patients were analyzed according to age (> or ≤4 years old) and to immunophenotype. Patients with other than Fab M7 presented (sporadic DS-AML) a dismal outcome with a significant difference (*p =* 0.000) in contrast to age (*p* = 0.149) (Figure 2). In the cox regression model, CK, immunophenotype and age were considered. The Fab M7 immunophenotype (classic ML-DS) was the strongest prognostic factor in univariate and multivariate analysis (*p* = 0.000) (Table 2).

#### 3.2.1. Sporadic DS-AML (Non-AMKL)

Six patients (14.6%) of DS-AML presented with immunophenotype other than Fab-M7: 1 Fab-M2, 4 Fab-M4, 1 Fab-M5. *GATA-1* and *FLT-3* mutations were analyzed in 2 patients by RT-PCR: none *FLT-3* and *GATA-1* mutations were found. One. Cytogenetic analysis was performed on 5 patients: isolated constitutional +21 was observed in 3 patients, one presented recurrent rearrangement t(8;21)(q22;q22)/*RUNX1-RUNX1T1* in combination with addition of chromosome 8 and deletion of crX, and the other patient presented del(6)(q21).

Three deaths-during induction were observed, all due to severe infection. One patient was a late-responder, dying later due to progressive disease. Only two patients achieved CR after AIE-block, one relapsed in bone marrow at 27 months of complete remission and one died in CR due to infection.

#### 3.2.2. Classical ML-DS (AMKL) and Comparison to Non-DS-AMKL

*GATA-1* mutations were found in 15 (93.8%) of 16 patients analyzed. Cytogenetic analysis was performed on 31 patients: isolated constitutional +21 was observed in 10 (32.3%) patients, being the addition of chromosome 8 the most frequent numerical alteration (19.4%) followed by i7(q) (9.7%), -cr7 (6.4%) and +cr22 (6.4%). CK was observed on 6 (19.3%) ML-DS.

When compared classical ML-DS and non-DS-AMKL there were statistically significant differences in clinical and demographic features regarding age and extra-medullary compromise (Table 1). As well as classical ML-DS, *FLT-3* mutations were not detected in the non-DS-AMKL population (0% and 2.3%, respectively). Patients showed comparable TMR (deaths-during induction and deaths in CR rates) (Table 3). Nevertheless, a statistically significant difference in CR rate associated with increased failure to achieve a CR after AIE-block and relapse rates were observed (Table 3). For classical ML-DS patients, the mean time from first CR to relapse was 4 (range: 1–7) months. None achieved a second CR. Deaths in CR were all due to infections.

The 5-years EFSp and OSp were 29.5 (SE 5.6, 95% IC 19.1–40.7)% for non-DS-AMKL vs. 72.8 (SE 7.7, 95% IC 57.6–88.6)% for classical ML-DS (*p* = 0.0001) and 5-years CIRp 52.9 (SE 7.4, 95% IC 39.8–67.8)% vs. 9.7 (SE 5.3, 95% IC 3.2–27.2)% respectively (*p* = 0.0001). (Figure 3). In the group of patients with classical ML-DS, the median time from first CR to relapse was 4.3 (range: 1–7) months vs. 6.2 (range: 1–19) months for non-DS-AMKL. Neither ML-DS nor non-DS-AMKL patients achieved a second remission after relapsing. As well as DS-AML, CK remained a significant prognostic factor for classical ML-DS patients. (Figure 3).

Analysis of administered treatment was done on 32/33 classical ML-DS patients that achieved CR after AIE-blocks: 7 (21.8%) received HAM-block, 30 (93.7%) received some type of consolidation block (AI/mHAM or Consolidation) and 12 (36.4%) Intensification-block. Considering the variability of treatment according to physician criteria, patients for better analysis were classified as receiving or not receiving at least one HDARA-C block (either HAM or Intensification-block). The 5-year EFSp was 88.9 (SE 10.5, 95% IC 43.3–98.4)% for patients who received HDARA-C (*n* = 13) vs. 73.7 (SE 10.1, 95% IC 47.9–88.1)% for those who did not receive HDARA-C (*n* = 19) (*p* = 0.172). In the HDARA-C group, one event was observed: a death in CR out of treatment; in the non-HDARA-C group, five events were observed: three relapses and 2 deaths in CR during treatment (Figure 4). Similar findings were observed when patients were divided according to receiving cumulative dosages of ARA-C > or < of 20 g/m^2^: the 5-year EFSp was 100% (*n* = 7) vs. 82.4% (SE 9.3, 95% IC 54.7–93.9)% (*n* = 17), (*p* = 0.251), respectively. The average of treatment duration for classical ML-DS was 15.1 (range: 1.6–22) months (Figure 4).

## 4. Discussion

TAM is a transient clonal myeloid proliferation with spontaneous regression without administration of chemotherapy a few months after diagnosis. The principle of TAM intervention, based upon this premise, is to reduce the severity of the symptoms rather than eradication of the clone [4]. Diagnosis criteria are still under debate [1,5,55]. In this report we describe 25 patients with TAM with demographics, hematological and clinical findings in concordance with the literature [1,2,3,4,5,6,8,15,16,17,18,23,28,29,30,31,32,33,56]. High age at onset was observed, probably related to diagnosis delay.

Even if there is no clear consensus regarding treatment indications and dosing, low doses of cytarabine are recommended for neonates with TAM and severe clinical impairment [1,2,4,5,6,8,15,16,17,18,19,23,56]. In this report, severe symptoms were observed in six (24%) patients: all disclosed hyperleukocytosis (>100,000/mm^3^), one associated with pericardial effusion and the other with liver failure and coagulopathy. Of the four (16%) patients who required cytoreduction, two had severe symptoms and two had leukocytosis (>50,000/mm^3^), none presented a later event. Overall mortality in our population was 16%, consistent with other reports (6–23%) [15,16,17,23]. None of the four deaths were related to TAM. Hepatic failure secondary to fibrosis has a central role in TAM with a fatal outcome [4,5,15,16,17,56]. In our setting, the patient with liver failure recovered hepatic function and remained in CR after early initiation of low-dose of cytarabine.

According to the literature, the majority of TAM (75–85%) only disclose the addition of chromosome 21 constitutional [15,16] and only 4% have a complex karyotype [22]. In this report, of 24 cytogenetic analyses performed, 84% presented only +21c, and no CK. Two patients developed AML, and only one had the cytogenetic analysis of both TAM and AML: an additional cr21 and deletion of cr7 were acquired on the blast clone in the AML phase.

The incidence of DS in our AML population was 5.7%. As expected, 95% were younger than four years of age [6,8,9,14,19,24,27,29,30,32,33,35,37,42,57,58,59], almost none (4.8%) had CNS involvement at diagnosis [29,30,33,34,37,58,59], only 6 (14.6%) presented with immunophenotype other than Fab M7 (sporadic AML-DS) [15,27,30,33,34,35,37,57,59] and *GATA-1* was mutated in 93.8% of ML-DS cases [42]. Also, only one patient of sporadic DS-AML aged 11 years old presented a favorable cytogenetic rearrangement [6,33,37,58,60]. As previously described, the incidence of chromosomal abnormalities was higher in AML than in TAM patients [22,61], being +8 the most frequent chromosome alteration = [30,33,34,35,37,41,57,59,60].

Refinement of AML therapy to optimize cure rates whilst minimizing toxicity is the major focus of modern treatment protocols. In our report, all DS-AML patients were treated according to BFM-AML trials without HSCT as part of first-line treatment and with a 2/3 dose reduction of anthracyclines as previously suggested [54]. No treatment failure related to cardiotoxicity was observed. All deaths-during induction and deaths in CR were due to sepsis complications, being infection-related mortality 24.4%. Although considerably higher than previously reported [24,27,29,30,31,32,33,34,37,39,57,59,62], a significant reduction was observed over time, probably associated with treatment modifications (2 HDARA-C intensification blocks in 1-AML90-BFM/HPG and 1-AML95-BFM/HPG vs. 1 HDARA-C re-induction and 1 HDARA-C consolidation block in 4-AML99-BFM/HPG^,^ and 12-AML07-BFM/HPG), as well as improvements on clinical support. When relapsed, DS-AML has a dismal prognosis [10,63,64]. According to the literature, the five-year CIR is 3–12.0% [24,29,31,34,57,58,62] and the 3-year OS rate is 8.5–34.3% [33,63,64]. In our report, the relapse rate was 13.9% and no patient achieved a second CR. For the total population of DS-AML, EFSp was 60.6 (8.2, 95% CI 42.8–74.4)%, lower than the reported value of around 80% [6,14,16,24,27,29,30,31,32,34,35,37,40,57,58,59,65], and considering outcome analysis was probably associated with TRM.

Different treatment groups have tried to identify adverse risk factors in order to optimize treatment regimens. As a general concept, age under 4-years old is the most validated good prognostic factor in AML-SD patients [10,31,32,37,58,59]. Complex karyotypes have been reported to be associated with an unfavorable therapy outcome in adult and childhood malignant myeloid disorders [66,67,68]. Worse outcomes related to CK were previously observed in DS-AML [23,41], although another group could not associate CK with worse OS, EFS or CIR [57]. In this report, CK, as well as non-megakaryoblastic immunophenotype and older age, were identified as poor prognostic factors. Interestingly, immunophenotype rather than age was the better predictor of dismal outcome.

Considering this finding, a separate comparison was made between classical ML-DS population and non-DS-AMKL patients enrolled in the same period. Although there is great concern about toxic mortality in DS patients, in this report, rates of TRM were similar between DS and non-DS children, and the better outcome of classical ML-DS is clearly associated with a remarkably low relapse rate when compared with non-DS-AMKL patients. Similar findings were observed in the AML-BFM and CCG study groups [31,37,59]. The classical ML-DS five-year EFSp was similar to that previously reported of around 82 (64.4–100%) [10,31,33,35,44,59,69].

The first major study disclosing that DS-AML patients represented a group with excellent outcomes was the Pediatric Oncology Group 8498 study [65]. Only a few studies compared outcome according cytarabine dose intensity.

According to the French Experience [27], patients treated with a low-dose cytarabine chemotherapy regimen presented an EFS of 45% compared to the 80.3% achieved for patients treated with a standard dose regimen (*p* < 0.01), making treatment intensity the only variant that significantly modified the probability of EFS associated with relapse rate reduction (*p* < 0.05) and not offset by increased mortality. The AAML0431 trial [34] identified MRD as a prognostic factor and the AAML1031 trial [41] evaluated a different treatment based on risk-stratification: cases with negative MRD (<0.05%) after the first course omitted HDARA-C; patients with positive MRD (>0.05%) received intensified therapy (equivalent to that used for non-DS AML). The observed EFS for MRD-negative patients was lower than expected for comparable MRD-negative patients whose treatment included a course of HDARA-C (AAML0431 trial [34]) (*p* = 0.0002), with a 9.6% relapse rate, and concluded that HDARA-C/E.coli asparaginase should be comprised in the treatment of ML-DS, regardless of MRD.

Expert review [10] supported that low-dose of ARA-C has limited evidence and is recommended only for those in whom more standard dosing of ARA-C is contraindicated. In our report, even if no significant difference in EFSp was observed according to doses of ARA-C, all relapses occurred in the non-HD-ARAC group and survival results showed a trend to achieve better results in the group of patients treated with HDARA-C.

## 5. Conclusions

We conclude that TAM represents the first manifestation of clonal myeloproliferative diseases in DS patients with good outcomes and rarely requires treatment. In those cases, with severe manifestations, early intervention is effective to prevent dismal outcomes. When DS patients develop AML, further karyotype aberrations are observed in blasts. Although the small sample size, in this study, we could associate their good outcome with the absence of CK, younger age, but especially with the AMKL immunophenotype. Classical ML-DS showed clearly better outcomes than DS-AMKL and the main responsible factor is the low relapse-rate in the first. Therefore, to achieve excellent treatment efficacy, it is important to avoid undertreatment, as this is likely to result in unnecessarily high rates of relapse. However, the augmented sensitivity to chemotherapy seen in DS patients must be kept in mind, as treatment-related mortality is still extremely high in this patient group. Despite the low number of patients, our data does not support the omission of HDAC in classical ML-DS. This small number of patients retrospective analysis might help in designing a more definitive, prospective, and randomized large multi-institutional study, in order to validate these findings.

Myeloproliferative disorders associated with DS are the result of a multistep clonal process, with emerging genetic events that co-operate with mutant *GATA1* clones. Cytogenetic profiling may help in further improving risk-based treatment for ML-DS patients.

## Figures and Tables

**Figure 1 cancers-14-03286-f001:**
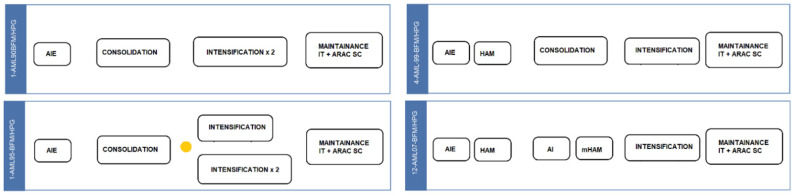
Overview of treatment schedules. AIE block: Cytarabine: 100 mg/m^2^, 24 h-IV infusion, days 1 and 2 Cytarabine: 100 mg/m^2^, 30 min-IV infusion, two doses, days 3–8 Idarubicin: 30 mg/m^2^, 4 h-IV infusion, days 3, 5, and 7 Etoposide: 150 mg/m^2^, 1 h-IV infusion, days 6, 7, and 8 DIT: doses according to patient’s age. Consolidation phase: Vincristine: 1.5 mg/m^2^, four doses, IV, weekly, weeks 1–4 Doxorubicin: 30 mg/m^2^, four doses, 6 h-IV infusion, weekly, weeks 1–4 Thioguanine: 60 mg/m^2^, seven doses per week, PO, daily, weeks 1–6 Cytarabine: 75 mg/m^2^, four doses per week, IV, weeks 1–6 Prednisone: 40 mg/m^2^, PO, daily, weeks 1–4 Cyclophosphamide: 500 mg/m^2^, two doses, IV, weekly, weeks 5 and 6 DIT: doses according to patient’s age, 4 doses, weeks 2, 4, 5 and 6. Intensification phase: HD Cytarabine: 3000 mg/m^2^, 3 h-IV infusion, two doses, days 1–3 Etoposide: 125 mg/m^2^, 1 h-IV infusion, days 2–5 DIT: doses according to patient’s age, day 1. Maintenance phase: Thioguanine: 40 mg/m^2^, daily, PO Cytarabine: 40 mg/m^2^, four doses per week, IV, monthly DIT: doses according to patient’s age, monthly during continuation phase. Second induction (HAM): HD Cytarabine: 3000 mg/m^2^, 3 h-IV infusion, two doses, days 1–3 Mitoxantrone: 10 mg/m^2^, 1 h-IV infusion, days 3 and 4 DIT: doses according to patient’s age, day 1. AI block (consolidation phase): Cytarabine: 500 mg/m^2^, 24 h-IV infusion, days 1–4 Idarubicin: 7 mg/m^2^, 1 h-IV infusion, days 3 and 5 DIT: doses according to patient’s age, days 0 and 6. hAM block (Consolidation phase): Cytarabine: 1000 mg/m^2^, 3 h-IV infusion, two doses, days 1–3 Mitoxantrone: 10 mg/m^2^, 1 h-IV infusion, days 3 and 4 DIT: doses according to patient’s age, days 6 and 15.

**Figure 2 cancers-14-03286-f002:**
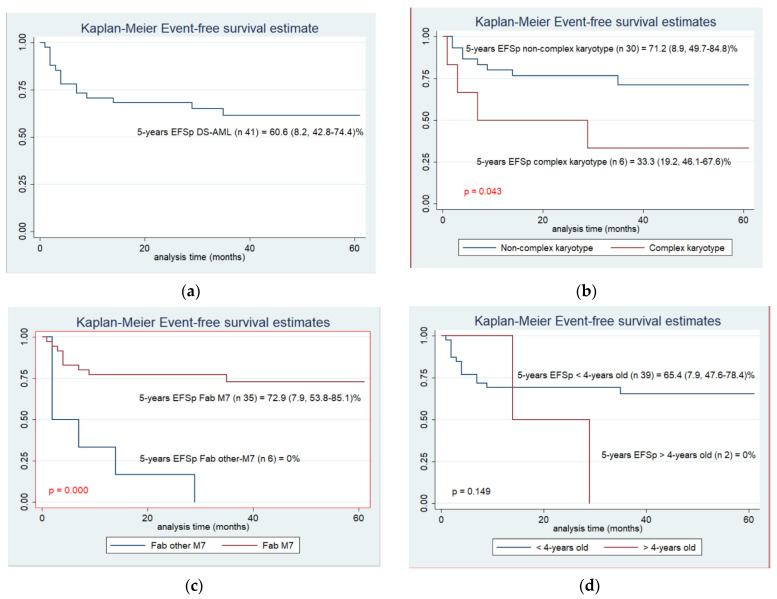
5-years EFSp for total DS-AML population and according to prognostic factors. (**a**) 5-years EFSp for total DS-AML population, (**b**) 5-years EFSp according to complex karyotype, (**c**) 5-years EFSp according to immunophenotype, (**d**) 5-years EFSp according to age > or < 4-years old.

**Figure 3 cancers-14-03286-f003:**
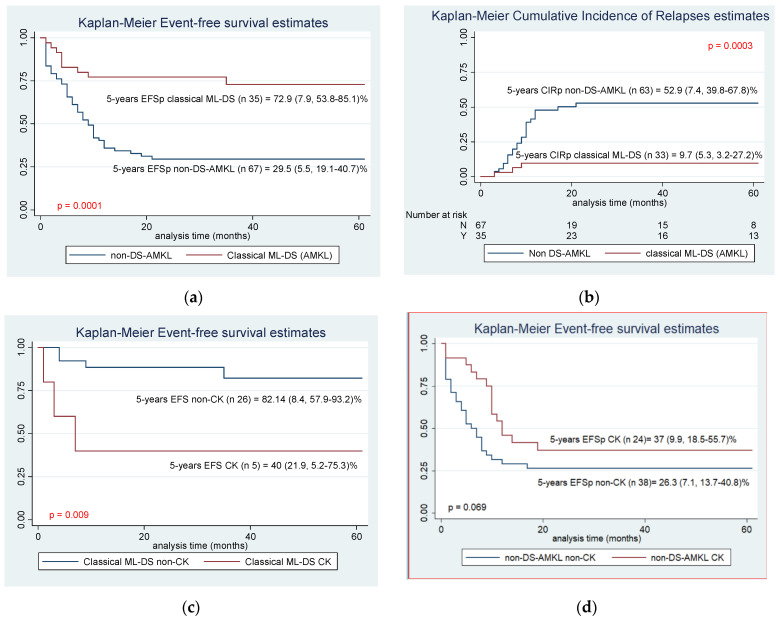
5-years EFSp and CIRp for total classical ML-DS (AMKL) and non-DS-AMKL population and complex karyotype. (**a**) 5-years EFSp for classical ML-DS (AMKL) and non-DS-AMKL population, (**b**) 5-years CIRp for classical ML-DS (AMKL) and non-DS-AMKL population, (**c**) classical ML-DS (AMKL) EFSp at 5-year according to complex karyotype, (**d**) non-DS-AMKL EFSp at 5-year according to complex karyotype.

**Figure 4 cancers-14-03286-f004:**
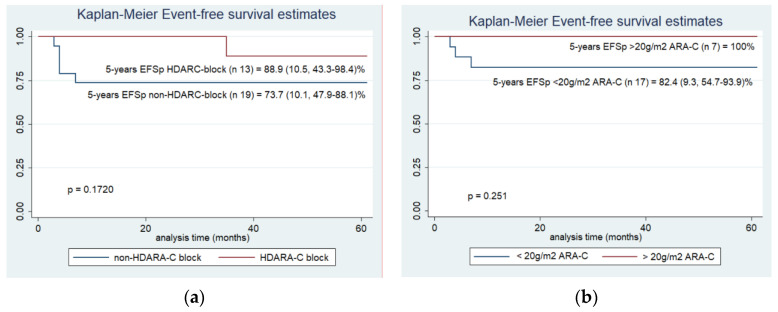
5-years EFSp for classical ML-DS (n31) who achieved CR, according cytarabine treatment dosis. (**a**) 5-years EFSp for classical ML-DS according HDARA-C block, (**b**) 5-years EFSp for classical ML-DS according receiving > or < 20 g/m^2^ cytarabine.

**Table 1 cancers-14-03286-t001:** Patients clinical, biological and demographic characterization of Down Syndrome–Acute Myeloid Leukemia (DS-AML).

Patients’ Characteristics	DS-AML (Total)		Classical ML-DS (AMKL)		Non-DS-AMKL		*p* Value *
	N 41	100%	N 35	100%	N 67	100%	
Female sex	22	53.6%	19	54.3%	29	43%	0.291
<4 years old	39	95.1%	35	100%	54	80.6%	**0.005**
WBC > 50,000/mm^3^	4	9.8%	2	5.7%	6	8.9%	0.550
E-MC	2	4.9%	1	2.8%	18	26.9%	**0.003**

Abbreviations: * *p* value difference between classical ML-DS (AMKL) and non-DS-AMKL groups. DS, Down syndrome; AML, Acute Myeloid Leukemia; ML-DS AMKL, acute Megakaryoblastic Leukemia; ML-DS, Myeloid Leukemia associated with Down Syndrome; WBC, White Blood Count; E-MC, Extra-medullary compromise. Bold, *p* value with statistically significant difference.

**Table 2 cancers-14-03286-t002:** Univariate and multivariate cox regression analysis for DS-AML outcome prognostic factors.

Prognostic Factors	Univariate Analysis			*p* Value *	Cox Regression Model Estimated Hazard Ratios (CI 95%)	*p* Value *
	Patients	Events	5-Year EFSp (SE)			
>4 years old <4 years old	**2**	**2**	0%	0.176	0.12 (0.12–1.16)	0.067
	39	13	65.4 (7.9)%		-	
CK non-CK	4	4	0%	0.057	4.02 (0.94–17.17)	0.06
	32	8	73.2 (8.3)%		-	
Fab other-M7 Fab M7	6	6	0%	**0.000**	20.13 (4.06–99.74)	**0.000**
	35	9	72.9 (7.9)%		-	

Abbreviations: * *p* value for the log-rank test on the difference between factor groups. CK, complex karyotype. Bold, *p* value with statistically significant difference.

**Table 3 cancers-14-03286-t003:** Outcome comparison between classical ML-DS (AMKL) and non-DS-AMKL.

Total Patients	Classical ML-DS (*n* = 35)		Non-DS-AMKL (*n* = 67)		*p* Value *
**Post-AIE BM CR**	**33**	94.3%	50	87.8%	**0.018**
Non/Late-response post-AIE	0	0%	9	13.4%	**0.018**
Deaths-during induction	2	5.7%	8	8.1%	0.170
**Deaths in CR**	4	11.40%	8	11.9%	0.939
Infectious	(4)	(100%)	(2)	(25%)	
ARDS	(0)	-	(2)	(25%)	
Related HSCT	(0)	-	(3)	(37.5%)	
UNK	(0)	-	(1)	(12.5%)	
**Relapses**	3	8.6%	25	37.3%	**0.002**
BM	(2)	(66.6%)	(22)	(88%)	
Combined	(1)	(33.3%)	(23)	(12%)	

Abbreviations: * *p* value difference between classical ML-DS (AMKL) and non-DS-AMKL groups. ML-DS, Myeloid Leukemia associated with Down Syndrome; DS, Down syndrome; AMKL, Acute Megakaryoblastic Leukemia; BM, Bone Marrow; CR, complete response; ARDS, Acute Respiratory Distress Syndrome; HSCT, Hematopoietic Stem Cell Transplantation; UNK, unknown. Bold, *p* value with statistically significant difference.

## Data Availability

The data presented in this study are available on request from the corresponding author. The data are not publicly available due to ethical reasons.

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
