# Peer review of "Clonal Myeloproliferative Disorders in Patients with Down Syndrome—Treatment and Outcome Results from an Institution in Argentina"

_cancers, 2022, doi:10.3390/cancers14133286_

Round 1
Reviewer 1 Report
In this article, the authors analyzed the balance between treatment intensity and treatment-related toxicity in DS-AML and DS-AMKL patients by a retrospective study in a single institute. The authors concluded that HDARA-C should not be omitted in DS-AMKL patients, although treatment-related mortality is still high. Since this type of study should be validated by a prospective and randomized design in multi-institutional study.
Author Response
Response to Reviewer 1 Comments
In this article, the authors analyzed the balance between treatment intensity and treatment-related toxicity in DS-AML and DS-AMKL patients by a retrospective study in a single institute. The authors concluded that HDARA-C should not be omitted in DS-AMKL patients, although treatment-related mortality is still high. Since this type of study should be validated by a prospective and randomized design in multi-institutional study.
Response 1. English language and style were reviewed. The information provided in the Introduction was expanded and Results were re-structured for better understanding. As the reviewer suggestions, in the conclusions, we state that this study should be validated by a prospective and randomized design in a multi-institutional study
Reviewer 2 Report
This is an extensive review of Down syndrome subjects seen at a single institution over a 10 year period. It focuses on those with specific hematologic findings and leukemia, and provides a useful look at these findings and their outcomes at this institution. An interesting aspect is the data on better outcomes in Down syndrome subjects versus non-Down syndrome subjects. This should be mentioned as confirmatory in comparison with other institutions. Minor issues: Line 11 in Abstract and lines 296-297 are awkward.
Author Response
Response to Reviewer 2 Comments
This is an extensive review of Down syndrome subjects seen at a single institution over a 10 year period. It focuses on those with specific hematologic findings and leukemia, and provides a useful look at these findings and their outcomes at this institution. An interesting aspect is the data on better outcomes in Down syndrome subjects versus non-Down syndrome subjects. This should be mentioned as confirmatory in comparison with other institutions. Minor issues: Line 11 in Abstract and lines 296-297 are awkward.
Response 1. English language and style were reviewed. Methods were modified according to reviewer 3 suggestions to better understand. As the reviewer suggests, in the conclusions we state that ML-DS had better outcomes versus non-Down syndrome subjects. Line 11 in the abstract and lines 296-297 were corrected to better understanding.
Reviewer 3 Report
In their manuscript, Pennella and colleagues collected and retrospectively analyzed cases of Down syndrome leukemia. This comprised cases with transient abnormal myelopoiesis, megakaryoblastic leukemia and sporadic AML, hence, a mixture of cases and biology.
- Please use the official nomenclature for myeloid leukemia associated with Down syndrome and transient abnormal myelopoiesis (according to WHO), including their abbreviations.
- While megakaryoblastic leukemia can evolve from TAM and therefore shares a common origin and pathobiology, sporadic AML is completely different. This difference is well known (see Hasle et al. 2015) and is mainly reflected by the different age (younger of older than 4 years), immunophenotype (megakaryoblastic leukemia vs other types of AML) and outcome (good vs. poor prognosis). Thus, although the collection of 6 DS-AML (non-megakaryoblastic) is interesting and should be reported, they should not be mixed with the AMKL cases. The section "DS-AML" should be splitted into classical ML-DS (AMKL) and sporadic DS-AML (non-AMKL). A Kaplan-Meier curve comparing the two could remain.
- Conclusions should be based on statistical power and significance. Both does not apply to the conclusion that HD-ARAC cannot be omitted from ML-DS therapy, which is currently tested in clinical studies. Moreover, it is unclear whether the HD-ARAC and non-HD-ARAC group are from different treatment protocols and therefore different time periods. Conounding variables were therefore not carefully excluded (inclduing multivariate analysis). Similarly, the conclusion that complex karyotype is a poor prognostic marker is based on 5 patients. Is the study powered to make such a conclusion?
- It is unclear, how many ML-DS patients received HAM. HAM is not recommended for ML-DS patients since the AML-BFM 98 trial.
- Many citations need a careful check. E.g. for the statement “persistent GATA1 mutant cells acquire additional mutations, most frequently in genes encoding members of the cohesin protein family, epigenetic regulators and signaling molecules” elven citations are referenced [1-4, 6, 8, 11, 12, 15, 16]. However, only two publications consider this connection (Yoshida et al. 2013 and Labuhn et al. 2019) of which the latter one is not even referenced.
- The material and methods sections (actually it should state patients and methods) needs to be revised. First, the treatment schedules over 30 years is unclear to the readership. I recommend including an overview of each block in each protocol. Second, cytogenetics and molecular genetic methods are scarce. Third, the diagnostic criteria need to be specifically stated. Especially for TAM, there is no consensus (see Tunstall et al. 2018 and Al-Kershi et al. 2021). For ML-DS, also cases with less then 20% blasts are considered (in contrast to AML). Fourth, the diagnostic criteria for AMKL need to be stated. Lastly, the definition of CR should be completely provided (recovery of neutrophils and thrombocytes?).
- Was cytogentics performed on all non-AMKL cases? Was the RUNX1-RUNX1T1 case among the non-AMKL ones?
- Discussion:
- First paragraph: the definition of TAM is under debate, which should at least be stated (see Tunstall et al. 2018 and Al-Kershi et al. 2021).
- The discussion is too long and should be should by half.
Author Response
Response to Reviewer 3 Comments
In their manuscript, Pennella and colleagues collected and retrospectively analyzed cases of Down syndrome leukemia. This comprised cases with transient abnormal myelopoiesis, megakaryoblastic leukemia and sporadic AML, hence, a mixture of cases and biology.
Point 1. Please use the official nomenclature for myeloid leukemia associated with Down syndrome and transient abnormal myelopoiesis (according to WHO), including their abbreviations.
Response 1. Nomenclature and abbreviations were modified according to WHO official nomenclature. Nomenclature was changed as well on figures and tables.
Point 2. While megakaryoblastic leukemia can evolve from TAM and therefore shares a common origin and pathobiology, sporadic AML is completely different. This difference is well known (see Hasle et al. 2015) and is mainly reflected by the different age (younger of older than 4 years), immunophenotype (megakaryoblastic leukemia vs other types of AML) and outcome (good vs. poor prognosis). Thus, although the collection of 6 DS-AML (non-megakaryoblastic) is interesting and should be reported, they should not be mixed with the AMKL cases. The section "DS-AML" should be splitted into classical ML-DS (AMKL) and sporadic DS-AML (non-AMKL). A Kaplan-Meier curve comparing the two could remain.
Response 2. According to the reviewer's suggestion, the section DS-AML was split into classical ML-DS (AMKL) and sporadic DS-AML (non-AMKL), and information was re-organized according to this division. In figure 1c, there is a Kaplan-Meier curve comparing the two populations.
Point 3. Conclusions should be based on statistical power and significance. Both does not apply to the conclusion that HD-ARAC cannot be omitted from ML-DS therapy, which is currently tested in clinical studies. Moreover, it is unclear whether the HD-ARAC and non-HD-ARAC group are from different treatment protocols and therefore different time periods. Conounding variables were therefore not carefully excluded (inclduing multivariate analysis). Similarly, the conclusion that complex karyotype is a poor prognostic marker is based on 5 patients. Is the study powered to make such a conclusion?
Response 3 The conclusions are based on the findings found in this study, clarifying the low number of patients and, according to the contributions made by reviewer 1, it is clarified that these findings should be validated by a larger multicenter prospective study.
Point 4. It is unclear, how many ML-DS patients received HAM. HAM is not recommended for ML-DS patients since the AML-BFM 98 trial.
Response 4. In line 122 of the “Materials and Methods” section, the recommendation to omit the HAM block is clarified and in line 294 of the “Results” section, it mentions that, however, 7 patients have received a HAM block throughout the period analyzed.
Point 5 .Many citations need a careful check. E.g. for the statement “persistent GATA1 mutant cells acquire additional mutations, most frequently in genes encoding members of the cohesin protein family, epigenetic regulators and signaling molecules” elven citations are referenced [1-4, 6, 8, 11, 12, 15, 16]. However, only two publications consider this connection (Yoshida et al. 2013 and Labuhn et al. 2019) of which the latter one is not even referenced.
Response 5. Citations were reorganized and carefully checked as reviewer 3 suggestions.
Point 6. The material and methods sections (actually it should state patients and methods) needs to be revised. First, the treatment schedules over 30 years is unclear to the readership. I recommend including an overview of each block in each protocol. Second, cytogenetics and molecular genetic methods are scarce. Third, the diagnostic criteria need to be specifically stated. Especially for TAM, there is no consensus (see Tunstall et al. 2018 and Al-Kershi et al. 2021). For ML-DS, also cases with less then 20% blasts are considered (in contrast to AML). Fourth, the diagnostic criteria for AMKL need to be stated. Lastly, the definition of CR should be completely provided (recovery of neutrophils and thrombocytes?).
Response 6. According to reviewer's suggestion a new figure was added (now Figure 1) with an overview of treatment protocols and treatment schedules. As well, cytogenetics and molecular genetic methods were described in greater detail. Diagnosis criteria of TAM, ML-DS, and AMKL were included. CR definition was completed.
Point 7. Was cytogentics performed on all non-AMKL cases? Was the RUNX1-RUNX1T1 case among the non-AMKL ones?
Response 7. Yes, cytogenetics was performed on 5 of 6 patients non-AMKL, and results are detailed in section 3.2.1 of “Results”. As well it is mentioned that the RUNX1-RUNX1T1 case is among the non-AMKL ones.
Point 8. Discussion:
- First paragraph: the definition of TAM is under debate, which should at least be stated (see Tunstall et al. 2018 and Al-Kershi et al. 2021).
- The discussion is too long and should be should by half
Response 8. The first paragraph of the discussion was modified to state that the TAM definition is under debate as the reviewer suggested. The discussion was shortened and some of the information was reorganized in the Introduction as reviewer 1 suggested.
Round 2
Reviewer 1 Report
The authors should present the results of randomized phase III study by prospectively performed in multicenters.
Reviewer 3 Report
The authors have greatly improved the manuscript by addressing the reviewer's concerns.